# Biogeography and Systematics of the Genus *Axyris* (Amaranthaceae s.l.)

**DOI:** 10.3390/plants11212873

**Published:** 2022-10-27

**Authors:** Alexander P. Sukhorukov, Valeriia D. Shiposha, Maria Kushunina, Maxim A. Zaika

**Affiliations:** 1Department of Higher Plants, Biological Faculty, Lomonosov Moscow State University, 119234 Moscow, Russia; 2Laboratory Herbarium (TK), Tomsk State University, Lenin Ave. 36, 634050 Tomsk, Russia; 3Department of Plant Physiology, Biological Faculty, Lomonosov Moscow State University, 119234 Moscow, Russia

**Keywords:** Amaranthaceae, Asia, *Axyris*, biogeography, molecular phylogeny, reproductive characteristic

## Abstract

*Axyris* is a small genus of six species with a disjunct geographic range. Five species are present in Siberia, Central Asia, the Himalayas, and Tibet, whereas *Axyris caucasica* has been registered in the Central Caucasus only. *Axyris* species diversity is the highest in the Altai Mountains (four spp.), followed by the Tian Shan and Pamir Mountains (three spp.), and the Himalayas and Tibet (two spp.). *Axyris sphaerosperma*, sometimes considered endemic to Southern Siberia, in fact has a disjunct range: it is present in the lowlands of Eastern Siberia and in the Altai, Tian Shan, and Pamir Mountains. It has also been found in Mongolia and China for the first time. An updated detailed distribution of *Axyris* in Siberia is presented on the basis of thorough herbarium revisions. One nuclear and three plastid markers were selected for phylogenetic analysis. Divergence times were estimated using a time-calibrated Bayesian approach. *Axyris* shows two major clades: an *Axyris amaranthoides* clade and a clade including the remaining species. The latter clade consists of two subclades (*A. sphaerosperma*/*A. caucasica* and *A. mira*/*A. prostrata* + *A. hybrida*). The crown age for *Axyris* dates back to the Early Pliocene (~5.11 mya, the Zanclean). The ancestral range of *Axyris* covers Southern Siberia, Mongolia, NW China, and the Tian Shan/Pamir Mountains, with extensions toward Eastern Siberia, the Himalayas/Tibet, and the Caucasus. Fruit and seed characteristics of *Axyris* are discussed with reference to the present phylogenetic results. Closely related *A. sphaerosperma* and *A. caucasica* have the thickest seed coat among all Chenopodiaceae, and these traits have probably evolved as adaptations to extremely low winter temperatures. This reproductive peculiarity may explain the disjunct range of *A. sphaerosperma*, which is restricted to harsh climatic conditions.

## 1. Introduction

*Axyris* L. is a small genus of Chenopodiaceae Vent. s.str. (Amaranthaceae Juss. s.l.) comprising six species [1]. All of them share an annual life form, stellate pubescence of the stem and leaves, unisexual flowers, and one-seeded indehiscent fruits, usually with ear-like outgrowths in their upper part originating from the pericarp [2]. Although the genus is easily recognizable in the field, identification at the species level has often led to confusion due to the absence of reliable diagnostic characteristics. Three species—*Axyris amaranthoides* L., *A. hybrida* L., and *A. sphaerosperma* Fisch. & C.A.Mey.—have often been misidentified owing to their upright stems, similar leaf shapes, and overlapping geographic ranges in some parts of temperate Asia. Diagnostic methods for each species were greatly improved only recently, and pubescence details of stems and leaves and specific features of fruits and seeds appear to be the key discriminatory traits [1,2]. All *Axyris* species produce both heterocarpous and heterospermous diaspores, and morphoanatomical traits of fruit types are now regarded as diagnostic at the specific level [1,2].

The main center of distribution of *Axyris* lies in Central Asia [3,4,5,6] and the Himalayas/Tibet [1]. A single species, *Axyris caucasica* (Somm. & Lev.) Lipsky, is present in the Greater Caucasus [3]. *Axyris* therefore demonstrates substantial geographical disjunction between two parts of its native range: temperate East Asia and the Greater Caucasus. Among all *Axyris* species, only *A. amaranthoides* has become an adventive alien with a scattered distribution in the Russian Far East [7], Eastern Siberia (Sakha Rep.; [8]), Eastern Europe [9], Fennoscandia [10], the North Caucasus [11], and North America [12].

*Axyris* is the type genus of the tribe Axyrideae G.Kadereit & Sukhor. [13]. The genus seems to be monophyletic, judging by two or three species included in a phylogenetic analysis [13,14,15], but infrageneric relationships have not been elaborated so far. The aims of the present study were (i) to reveal the main diversity centers of *Axyris*, (ii) to construct an *Axyris* phylogeny comprising all the species of the genus and to revise the systematics of the genus, and (iii) to trace biogeographical history and radiation of *Axyris* on the basis of the phylogenetic analysis.

## 2. Results

### 2.1. Species Distribution of Axyris in Asia

We revised the distribution of *Axyris* in Siberia, and in particular, we found that the range of *A. sphaerosperma* is drastically different from the one previously reported (Figure 1, Figure 2, Figure 3 and Figure 4). Based on our recent results and the previous study [1], the highest species diversity of *Axyris* is seen in Southern Siberia, Mongolia, and NW China (Figure 5), where four species are present (*A. amaranthoides, A. hybrida, A. prostrata,* and *A. sphaerosperma*), followed by the Tian Shan and Pamir Mountains (*A. hybrida, A. prostrata*, and *A. sphaerosperma*) and the Himalayas/Tibet (*A. mira* and *A. prostrata*). One native species each is present in Eastern Siberia (Sakha [Yakutiya] Republic) and in the Greater Caucasus: *A. sphaerosperma* and *A. caucasica*, respectively.

### 2.2. Dated Molecular Phylogeny of Axyris

The combined dataset of all four markers (ITS, *rbcL* coding gene, *atpB*-*rbcL* and *trnL*-*F* intergenic spacers) comprises 2588 aligned bp and 21 accessions. ML and Bayesian analyses revealed identical topologies. *Axyris* is monophyletic with high support and it is recovered as a sister clade to *Krascheninnikovia* and *Ceratocarpus* (Figure 6; BSL 100; PP 1). *Axyris amaranthoides* is resolved as sister to the rest of the species (BSL 91; PP 1). *Axyris caucasica* and *A. sphaerosperma* form a clade (BSL 96; PP 1) sister to a well-supported clade (BSL 82; PP 0.95) consisting of *A. mira*/*A. prostrata* + *A. hybrida*.

The diversification of *Axyris* started in the late Eocene ca. 25.65 mya (stem age, 95% HPD: 36.3–14.97). The crown age of *Axyris* dates back to ca. 5.11 mya (95% HPD: 10.91–2.82 mya) which suggests that the genus originated during the early Pliocene (the Zanclean stage).

### 2.3. Biogeographic Analyses and Radiation of Axyris

The ancestral area for the crown node of *Axyris* could be estimated for the regions A (Southern Siberia, Mongolia, NW China) and B (Tian Shan, Pamir Mountains) (AB: *p* = 0.55, ABE: *p* = 0.13, A: *p* = 0.11; ABD: *p* = 0.10; Figure 7). It is likely that the ancestral *Axyris* domain can be linked to the geographic area where four species of the genus are present today (Figure 5; marked with orange). For all other nodes (except node 9, ABE: *p* = 59, ABCE: *p* = 30), the likely ancestral domain is also confined to AB areas with varying probabilities (Figure 7, see Legend). Further dispersal of *Axyris* proceeded towards the Himalayas/Tibet (Region D: *A. mira* and *A. prostrata*), which was colonized once, and then Eastern Siberia (Sakha Republic, Region E) (*A. sphaerosperma*) and the Caucasus, Region C (*A. caucasica*).

## 3. Discussion

### 3.1. The Distribution of Axyris in Siberia

Four out of six *Axyris* species—namely *A. amaranthoides*, *A. hybrida*, *A. prostrata*, and *A. sphaerosperma*—are native to Siberia [3]. According to the latest examination of the genus in Siberia [16], *A. amaranthoides* is widespread in Southern Siberia, with a few records in Northern and Eastern Siberia; *A. hybrida* mostly occurs in Southern Siberia with scattered findings in Eastern Siberia; and *A. prostrata* is present in Southern Siberia. No distribution map of *Axyris sphaerosperma* was provided by M. Lomonosova [16], but several findings in Altai and Tyva Republics were reported, as was one finding in the city of Yakutsk (Sakha Republic). 

According to our investigations, *A. amaranthoides* is a common plant in most of Siberia, but its records in the northern and northeastern parts of the region represent recent migrations as a consequence of human activities. In Eastern Siberia, it is more frequently found in settlements and along main transport routes, especially in the Lena River basin. *A. hybrida* occurs across Southern Siberia, mostly in the Altai and Sayan Mountains but also in the adjacent lowlands. The findings of *A. hybrida* in Sakha Republic (Eastern Siberia) reported by the authors of refs. [16] and [8] are not confirmed, and these records in fact belong to *A. amaranthoides*. As stated by M. Lomonosova [16] and confirmed by us, the geographic range of *A. prostrata* is mostly confined to the Altai and Sayan Mountains, with scattered records in the uplands of Baikalia and Krasnoyarsk Krai.

### 3.2. A Geographical Puzzle concerning A. sphaerosperma

Out of all *Axyris* species, only *A. sphaerosperma* possesses a disjunct range, with two fragments (Figure 4) reported here for the first time. The first (main) fragment is located in the Altai and Sayan Mountains, with extensions to the Tian Shan, Pamir, and North Himalayas [1,3,5]. Here we also add the first but expected records for Mongolia (Mongolian Altai, Bayan-Ulgii prov., Tavan-Bogd Mts., Tsagaan-Gol River basin, 4 August 2001, I. Krasnoborov, A. Shmakov, and D. German 82 (NS0009424); and China (Irenchabirga, 9000 ft, 10 September 1879, A. Regel [LE]; Zaisan expedition [Xinjiang prov., Habahe county], Kobuk River basin, 20 July 1914, V. Sapozhnikov s.n., all as *A. amaranthoides* [LE]) that have not been mentioned earlier (e.g., [6,17,18]). These new findings were made close to those in Altai Republic (Southern Siberia, Russia). In the main fragment of its range, *A. sphaerosperma* occurs at high altitudes, between 1500 and 3000 m a.s.l. [19]. The second fragment is located in the lowlands of Eastern Siberia (Sakha Republic). Both areas are characterized by extremely low winter temperatures and a short vegetation period lasting from mid-May to mid-September. The reasons for such a fragmentary range of *A. sphaerosperma* are unclear. Nonetheless, it should be noted that *A. sphaerosperma* is well adapted to the harsh climatic conditions by forming a persistent soil seed bank. Although all *Axyris* species have heteromorphic fruits and seeds with different longevity, some fruits of *A. sphaerosperma* and *A. caucasica* have sclereids and contain seeds with a very hard and thick (up to 100–115 μm) seed coat [2]. Such thickness of the seed coat is exceptional for Chenopodiaceae [9,20] and is a good example of physical dormancy in species growing under harsh climatic conditions.

### 3.3. The Origin of Axyris

The origin of this genus is probably connected with the orographic and climatic changes in the late Miocene/early Pliocene, which are manifested in renewed tectonism and further aridification of the lowlands in South Siberia and Tian-Shan [21]. The same region of origin, for example, is reported for *Krascheninnikovia ceratoides* (L.) Gueldenst. [22,23], a species also belonging to Axyrideae [13]. It is still not clear whether the origin of *Axyris* and *Krascheninnikovia* is linked to high-altitude or lowland steppes, but it is thought that the late Miocene and early Pliocene are the time scales where a continent-wide restructuring of the distribution of landscape-forming elements was taking place, including a new zonal structure component: steppe formation [24]. Our time-calibrated tree coincides with previous study [25]. Divergence time between *Axyris* and *Krascheninnikovia* + *Ceratocarpus* group is 26.5 mya and between *Krascheninnikovia* and *Ceratocarpus* is 16.4 mya, which generally corresponds with our data (25.6 and 15.9 mya, respectively). 

According to the altitudinal gradient, species of *Axyris* can be subdivided into two main groups: (1) predominantly lowland species (only *A. amaranthoides*) and (2) predominantly mountain species (all the other taxa), sometimes penetrating onto the lowlands (*A. hybrida*, *A*. *prostrata*, and *A. sphaerosperma*). None of them can be classified as desert plants, and there is evidently a gap in the distribution of the genus in the Taklamakan desert. A considerable gap between the main distribution area of the genus located in temperate East Asia and a small fragment in the Greater Caucasus (*A. caucasica*) can be explained by climatic changes, including Paleo-Caspian Transgression in the lowlands of Kazakhstan during the Pliocene and Pleistocene [26]. As stated earlier [1], one type of reproductive diaspores of both *A. sphaerosperma* and *A. caucasica* is very thick and provides high seed longevity. The reproductive strategy of these species is connected with the adaptation to extremely low winter temperatures. We suppose that the precursors of these species were present in the Pliocene and early Pleistocene in the lowlands of the Aralo-Caspian floristic province reaching the Caucasus, the areas where permafrost never disappeared during warm phases [27]. Decreasing permafrost led to the desertification of the landscape [28] and therefore could induce the disappearance of cold-adapted species in the lowlands and their isolation in the areas with much colder winter temperatures. 

### 3.4. Systematics of Axyris

This topic is elaborated based on a molecular phylogeny for the first time here. As proposed earlier [1,2], carpological data fully support the new systematic subdivision of *Axyris* into three sections. 

Gen. Axyris L., Sp. Pl.: 979 (1753).

Monoecious annuals covered with stellate hairs sometimes intermixed with simple multicellular hairs. Leaves short- or long-petiolate; blades ovate, oblong, spatulate, or lanceolate, entire, rarely undulate. Male flowers arranged in terminal spike-like inflorescences up to 8 cm long, with minute perianths of five free hyaline segments and with 2–5 stamens; female flowers located in bract axils, with five prominent hyaline perianth segments (two of which are erroneously called bracteoles). Fruits always dimorphic (heterocarpous); pericarp tightly adhering to the seed coat, usually with ear-like appendages at the apex of the fruit. Seeds also dimorphic (with thick and thin testal layer of the seed coat). Embryo vertical, horseshoe-shaped (in flattened fruits), or annular (in spheroidal fruits); perisperm present.

Six species in Eurasia, predominantly in Central Asia; one (*A. amaranthoides*) grows as an alien in many parts of Europe and North America. 

Lectotype (Jonsell and Jarvis in [29]): *Axyris amaranthoides* L.

#### The New Sectional Subdivision of the Genus

Key to the sections

Stellate hairs with short and long rays; black fruits spheroidal, brown fruits compressed—*Axyris* sect. *Sphaerospermae*
-Stellate hairs with short rays; fruits of both types compressed … 2
Ear-like pericarp appendages touching each other; fruits without concentric ridges—*Axyris* sect. *Axyris*
-Ear-like pericarp appendages not touching each other; black fruits with concentric ridges—*Axyris* sect. *Hybridae*



***Axyris* sect. *Axyris***


Stellate hairs with short or slightly elongated rays; fruits of both types compressed; black fruits smooth (without concentric sculpture) with small apical appendages touching each other, seed coat 30–45 (55) μm; brown fruits with large ear-like appendages also touching each other, seed coat 20–25 μm.

One species, *Axyris amaranthoides* L. (type species of the genus).

**1.** ***Axyris amaranthoides*** L., Sp. Pl.: 979 (1753).

Lectotype (Jonsell and Jarvis in [29]): Herb. Linn. 1101.4 (LINN!). Image available at: http://linnean-online.org/10713/ (accessed on 2 June 2022)

= *Axyris prostrata* L. var. *diffusa* Fenzl in Ledeb., Fl. Ross. 3: 714 (1849).

Lectotype (designated here): “herb. Ledebour” (LE01019599!) upper twig at fruiting stage.

= *Axyris amaranthoides* L. var. *dentata* A.I.Baranov, Zap. Kharbin. Obsch. Estestvoisp. Etnogr. 12: 35 (1954).

≡ *Axyris amaranthoides* L. f. *dentata* (A.I.Baranov) Kitag., Neolin. Fl. Manshur.: 248 (1979). 

Type: [China] prov. Cheilungkiang [Heilongjang prov.] ad ostium fl. Argun, pagus Strelka, in ruderatis, 29 July 1950, A. Baranov [type collection not designated]. 

Note. A variety with slightly undulate leaves. Such plants are sometimes found in different parts of the species range.

= *Axyris koreana* Nakai, J. Jap. Bot. 15(9): 525 (1939). 

Type: Korea, prov. Kannan, secus lacum artificialem tractus Tyôsin, 7 August 1938, S. Zen *5* (TI—image seen!). 


***Axyris* sect. *Hybridae* Sukhor., sect. nov.**


Stellate hairs with short or slightly elongated (*A. prostrata*) rays; fruits of both types compressed; black fruits with concentric ridges or rugose, with small apical appendages not touching each other, seed coat 25–50(65) μm; brown fruits with small or hardly noticeable appendages also not touching each other, seed coat 7–15 μm.

Type species: *Axyris hybrida* L.

Three species: *Axyris hybrida* L., *A. prostrata* L., and *A. mira* Sukhor., in Central Asia, Southern Siberia, and the Himalayas/Tibet. Two of them are present in Russia.

**2.** ***Axyris hybrida*** L., Sp. Pl.: 980 (1753).

Lectotype (Sukhorukov, Fedd. Repert. 116(3–4): 175 (2005)): Herb. Linn. 1101.5 (LINN!). 

= *Axyris amaranthoides* L. var. *stricta* Fenzl in Ledeb., Fl. Ross. 3: 713 (1851).

Lectotype (designated here): “Herb. Ledebour 8711” LE!, left-handed specimen).

= *Axyris hybrida* L. var. *eravinensis* Peshkova in Malyshev and Peshkova, Fl. Tsentrl. Sibiri 1: 299 (1979).

Type: [Russia] Buryat Republic, Eravninsky distr., in loco dicto Barun-Uldurga, ad viam, 23 July 1953, Takhistova s.n. (NSK—image seen!).

**3.** ***Axyris prostrata*** L., Sp. Pl.: 980 (1753).

Lectotype (Sukhorukov, Fedd. Repert. 116(3–4): 175 (2005)): Herb. Linn. 1101.6 (LINN!). 

= *Axyris prostrata* L. var. *latifolia* Fenzl in Ledeb., Fl. Ross. 3: 714 (1849).

Lectotype (designated here): “Altai. 1826 [fr.], № 871.2.2. Herb. Ledebour”. (LE01019600!).

= *Axyris pamirica* B.Fedsch., Acta Hort. Petrop. 24(3): 342 (1905).

≡ *Axyris prostrata* L. var *pamirica* B.Fedtsch., Acta Hort. Petrop. 24(3): 342 (1905).

Holotype: [Tajikistan, Gorny Badakhshan] Pamir, Gorumdy, 27 August 1904, B.A. Fedchenko s.n. (LE!).

**4.** ***Axyris mira*** Sukhor., Willdenowia 41(1): 76 (2011).

Holotype: [India, Uttarakhand State] Kumaon, Milam glacier, 12,500 ft above the sea, 28 August 1848, R. Strachey and J.E. Winterbottom 2 (LE!).


***Axyris* sect. *Sphaerospermae* Sukhor., sect. nov.**


Stellate hairs with short and long rays; black fruits spheroidal, brown fruits compressed; black fruits smooth or with ± longitudinally striate sculpture (concentric ridges absent), apical appendages small or almost unnoticeable, not touching each other, seed coat (40)50–90(115) μm; brown fruits with small or hardly noticeable appendages also not touching each other, seed coat 12–20(25) μm.

Type species: *Axyris sphaerosperma* Fisch. & C.A.Mey.

Two species: *Axyris sphaerosperma* (in Central Asia, Sakha Republic, Pamir/Tian Shan Mts., and North Himalayas) and *A. caucasica* (Somm. & Lev.) Lipsky (in the Greater Caucasus).

**5.** ***Axyris sphaerosperma*** Fisch. & C.A.Mey. in Fisch., C.A.Mey. & Ave-Lall., Index Sem. Hort. Petrop. 6: 46 (1839).

Lectotype (Sukhorukov, Fedd. Repert. 116(3–4): 175 (2005)): [Russia, Altai Republic] In regione orientali fl. Altaicae, Tschuja. [anno] 1839, [anonymous] (LE01019601!).

**6.** ***Axyris caucasica*** (Somm. & Lev.) Lipsky, Fl. Cauc.: 430 (1899).

≡ *Axyris sphaerosperma* Fisch. & C.A.Mey. var. *caucasica* Somm. & Lev., Acta Hort. Petrop. 13(2): 196 (1894).

Lectotype (designated here): [Russia, North Caucasus, Karachay-Cherkessia Rep.] in jugo Tieberdinski pereval dicto, inter flumina Tieberda et Do-ut [Dout], ditionis Kuban, 1400 m, 1 September 1890, S. Sommier and E. Levier s.n. (LE!).

## 4. Materials and Methods

The taxonomic revision of the herbarium material was conducted at ABGI, G, LE, LECB, MHA, MSK, MSKU, MW, MWG, MOSP, NS, NSK, PE, PVB, RV, RWBG, TK, TLT, VOR, and WIR. The field investigations were carried out by the first author (A.P.S.) in Southern Siberia and the Far East (Amur Oblast). The data on the distribution of *A. mira* and *A. prostrata* in Himalaya and Tibet were taken from ref. [1]. All records for each species are given in Appendix B. Distribution maps are based on the specimens cited and were prepared using the SimpleMappr online tool (http://www.simplemappr.net, accessed on 4 May 2022). 

### 4.1. Sampling and DNA Extraction, Amplification, and Sequencing

Sixty accession numbers were included in the phylogenetic analyses, representing all six *Axyris* species, as well as 24 accession numbers as outgroups from Chenopodiaceae/Amaranthaceae; the samples are listed in Table 1.

DNA was extracted from 5–10 mg of dried leaf samples from the herbarium specimens by means of the DNeasy Plant Mini Kit (Qiagen, Valencia, CA, USA). One nuclear (the nuclear ribosomal internal transcribed spacer, nrITS) and three plastid markers (protein-coding gene *rbcL* and two intergenic spacers: *atpB*-*rbcL* and *trnL*-*trnF*) were selected for the phylogenetic analysis. 

PCRs were carried out in Thermal Cycler T100 (Bio-Rad, USA) using the primers and cycler programs listed in Table 2.

PCRs for all primers were conducted in 25 μL reaction mixtures consisting of 5 μL of DNA (10 ng/μL), 1 μL of each primer, 0.5 μL of Encyclo polymerase (Evrogen, Russia), 0.5 μL of 50× dNTP, 5 μL of a 10× Encyclo buffer, and 14.5 μL of mQ. 

PCR products were purified with the Cleanup Mini BC023S Kit (Evrogen, Russia). Sanger sequencing was carried out at Evrogen JSC (Moscow, Russia); the sequencing primers were the same as the amplification primers.

### 4.2. Sequence Alignment, Phylogenetic Analyses, and Molecular Dating

Sequences were aligned with MAFFT v.7 at default parameters [35], and the alignment was adjusted manually in PhyDe v.0.9971 [36]. Gaps were treated as missing data during the phylogenetic inference.

Two separate analyses were performed on nuclear and plastid DNA datasets via Bayesian inference (BI) and ML. According to the Akaike information criterion (AIC), the best-fitting model was the GTR + G model for the plastid dataset and the GTR + G + I model for the nuclear dataset, respectively. For the ML analyses, we employed RAxML v.8 [37]. Bootstrap analyses were conducted with 2500 replicates for ML. Due to the lack of statistically significant incongruence between nuclear and plastid trees (Appendix A), a combined sequence matrix was compiled for further analysis.

Divergence times for *Axyris* taxa were estimated using a Bayesian uncorrelated log-normal relaxed clock under a birth–death speciation process [38] for the combined dataset. We selected a normal distribution for the secondary calibration with a mean of 59.2 and standard deviation of 4.3, equivalent to the 95% HPD estimate from ref. [39] for the crown of Chenopodiaceae s.str. Bayesian analyses were conducted in BEAST v.2.6.7 [40]. Four Markov Chain Monte Carlo analyses with four chains were run for 20 million generations for every dataset, with sampling every 20,000 generations. Output log files were analyzed by means of TRACER v.1.6 [41] to assess convergence and ESS of all parameters; 15% of samples were removed as burn-in prior to combining the independent runs with the help of LOGCOMBINER v.2.6.7 [40]. The maximum clade credibility tree was generated using TREEANNOTATOR v.2.4.5 [40].

### 4.3. Biogeographical Analysis

Geographic distributions of all the studied species were inferred from herbarium specimens. Eight large geographic regions reflecting the worldwide distribution of Chenopodiaceae s. str. were coded as follows: A = Southern Siberia (Altai and Sayan Mountains and adjacent areas; Asiatic Russia), Mongolia, NW China (Xinjiang); B = Tian Shan and Pamir Mountains (Kyrgyzstan, East Kazakhstan, East Uzbekistan, Tajikistan); C = Greater Caucasus (Russia, Georgia, Azerbaijan); D = Himalayas/Tibet (Xizang, Qinghai and Sichuan provinces of China, Nepal, Bhutan, Jammu and Kashmir, Uttarakhand and Himachal Pradesh States of India); E = Eastern Siberia (Sakha Rep.); F = North America.

The BI gene trees were pruned to remove all duplicate accessions using the drop.tip function in the ape package [42]. Ancestral range estimation was conducted by means of the time-calibrated tree representing six species of *Axyris* with only one accession per species using “BioGeoBEARS” [43,44] in R v.4.1.3 [45]. The coded geographic data are displayed in Table 3.

We ran the analysis in accordance with a dispersal–extinction–cladogenesis model (DEC model), dispersal–vicariance model (DIVALIKE model), or BAYAREA model (BAYAREALIKE model) and examined a second run adding parameter “j” (founder-event speciation) for each biogeographic model. Out of the six models explored in this study, the DEC + J model was the best fit judging by the AIC and likelihood ratio test results (Table 4). The analyses were unconstrained (without possible dispersal routes or ancestral areas assumed a priori).

We allowed the inferred ancestor to occupy a maximum of three areas corresponding to the largest number of areas occupied by any extant species.

## Figures and Tables

**Figure 1 plants-11-02873-f001:**
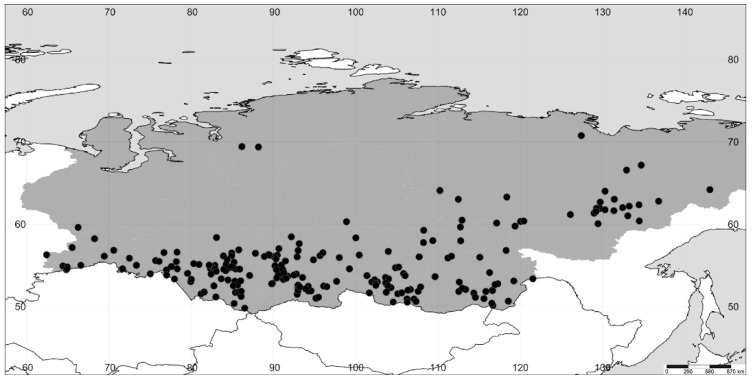
The distribution of *A. amaranthoides* in Siberia (the region itself is colored gray).

**Figure 2 plants-11-02873-f002:**
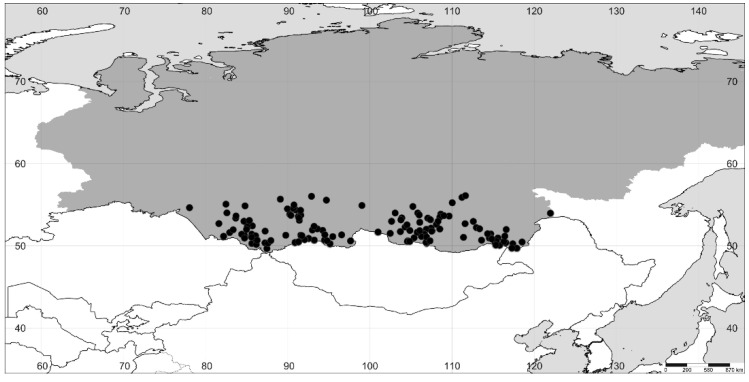
The distribution of *A. hybrida* in Siberia (the region itself is highlighted in gray).

**Figure 3 plants-11-02873-f003:**
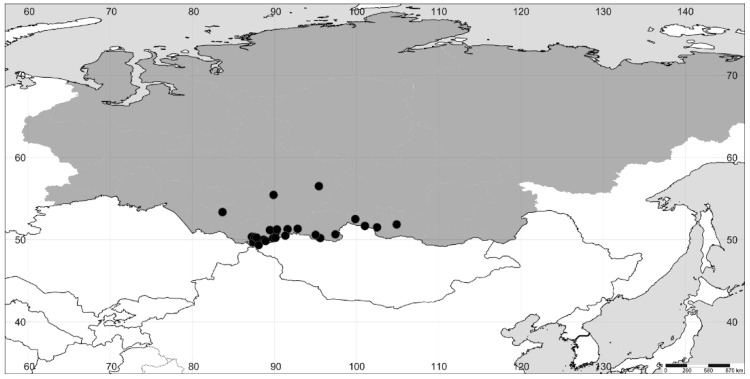
The distribution of *A. prostrata* in Siberia (the region itself is colored gray).

**Figure 4 plants-11-02873-f004:**
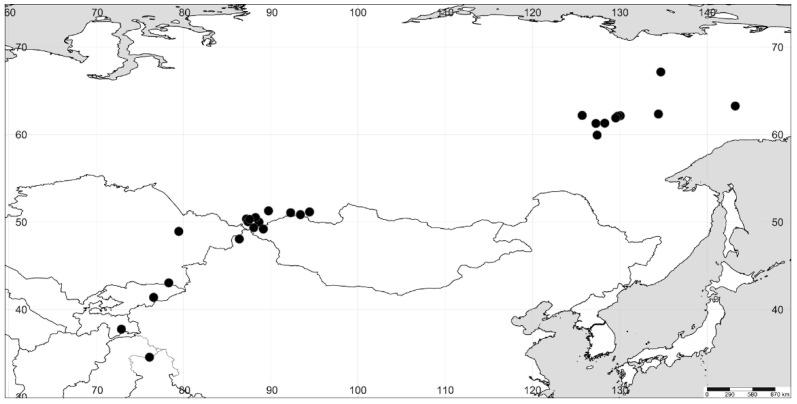
The distribution of *A. sphaerosperma*.

**Figure 5 plants-11-02873-f005:**
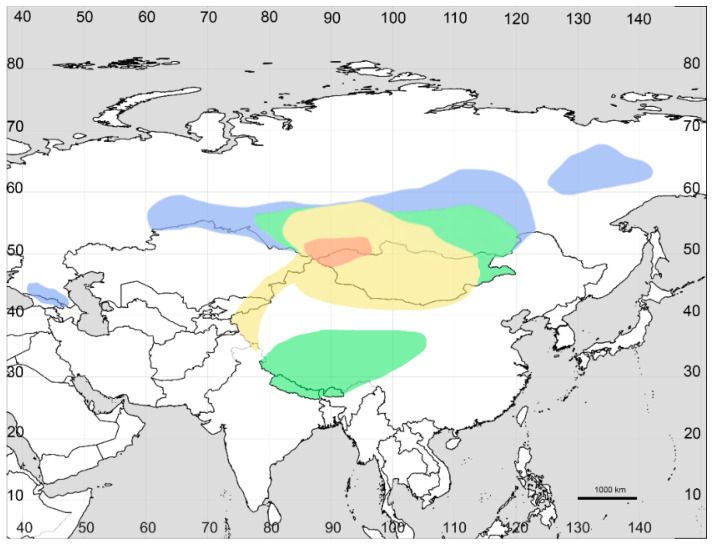
Species diversity of *Axyris*. Colored areas: orange, four species; yellow, three species; green, two species; blue, one species.

**Figure 6 plants-11-02873-f006:**
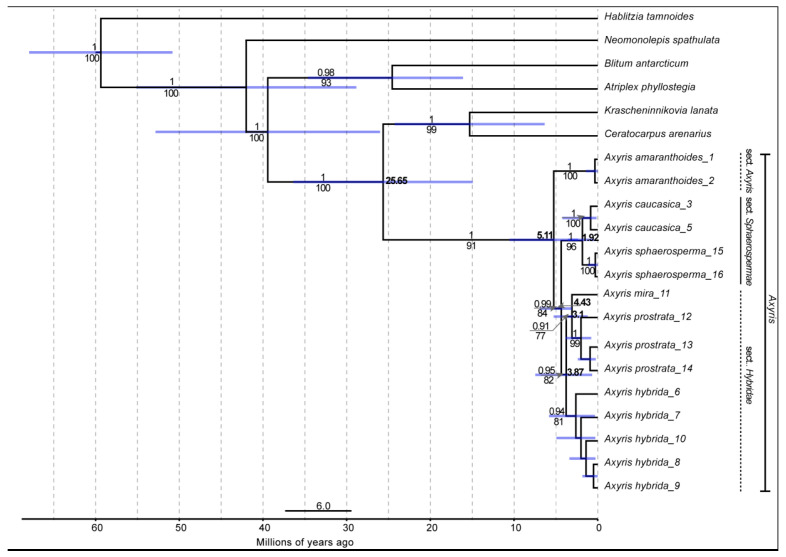
The maximum clade credibility tree of *Axyris* obtained by the BEAST2 analysis was subjected to secondary calibrations (see Methods). Posterior probabilities resulting from the Bayesian analysis are indicated above branches (only values ≥ 0.9), and the numbers below branches refer to bootstrap values resulting from the ML analysis (only values ≥ 70). Mean divergence times (values at some nodes) are shown with their 95% HPD (blue bars).

**Figure 7 plants-11-02873-f007:**
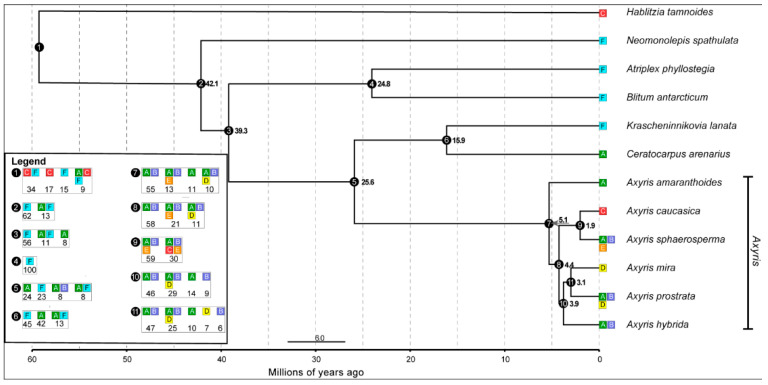
The time-calibrated tree of six *Axyris* taxa, as generated in BEAST2 allowing one accession per species. The ancestral area analysis was conducted by means of BioGeoBEARS in R v.3.3.2. Coding of biogeographical areas: A: Southern Siberia, Mongolia, and NW China; B: Tian-Shan and Pamir; C: Greater Caucasus; D: Himalayas/Tibet; E: Eastern Siberia (Sakha Rep.); F: North America.

**Table 1 plants-11-02873-t001:** Voucher information and GenBank accession numbers for the species of *Axyris* and outgroups included in the phylogenetic analysis.

Species	Voucher	ITS	*rbcL*	*atpB-rbcL*	*trnL-F*
*Axyris amaranthoides*_1	Russia, Tyva Rep., Piy-Khemsky distr., nr Turan town, 11 August 2002, V.V. Nikitin et al. 1011 (LE)	ON775477	ON783748	ON783763	ON783778
*Axyris amaranthoides*_2	Russia, Amur prov., Blagoveschensk town, 50.270989, 127.565183, 7 September 2020, A. Sukhorukov 52 (MW)	ON775478	ON783749	ON783764	ON783779
*Axyris caucasica*_3	Russia, Karachaevo-Cherkessiya, Karachaevsky distr., nr Khurzuk vill., 11 August 2015, A.S. Zernov 7937 (MW0678817)	ON775479	ON783750	ON783765	ON783780
*Axyris caucasica*_5	Russia, Kabardino-Balkar Rep., Malka River, 13 July 2010, Bondarenko (LE)	ON775480	ON783751	ON783766	ON783781
*Axyris hybrida*_6	China, Xinjiang, [without date and collector], (XJBI00071453)	ON775481	ON783752	ON783767	ON783782
*Axyris hybrida*_7	Russia, Khakassiya Rep., Altaisky distr., Izykhslie Kopi vill., 21 August 2011, I. Shantser and N. Stepanova 111 (MHA)	ON775482	ON783753	ON783768	ON783783
*Axyris hybrida*_8	Russia, Buryat Rep., Baikal Nature Reserve, 21 August 2017, N.S. Gamova 2590 (MW0163600)	ON775483	ON783754	ON783769	ON783784
*Axyris hybrida*_9	Russia, Altai Rep., Chemalsky distr., nr Elanda vill., 13 August 1985, I. Pshenichnaya and G. Liventsova s.n. (MW0058817)	ON775484	ON783755	ON783770	ON783785
*Axyris hybrida*_10	Russia, Irkutsk prov., Olkhovsky distr., 31 July 2010, S.G. Kazanovsky 21796 (MW0160613)	ON775485	ON783756	ON783771	ON783786
*Axyris mira*_11	Nepal, Mugu Karnali valley, 16 August 1952, O. Polunin et al. 5251 (LE);	ON775486	ON783757	ON783772	ON783787
*Axyris prostrata*_12	Russia, Altai Rep., Kosh-Agach, Kurayskaya steppe, 24 August 2003, M. Knyazev s.n. (LE)	ON775487	ON783758	ON783773	ON783788
*Axyris prostrata*_13	Russia, Altai Rep., Kosh-Agach, 12 July 1990, R. Kamelin et al. s.n. (LE)	ON775488	ON783759	ON783774	ON783789
*Axyris prostrata*_14	Russia, Tyva Rep., Bay-Tayginsky distr., nr Kara-khol Lake, 1460 m, 23 July 1976, I. Krasnoborov et al. 498 (LE)	ON775489	ON783760	ON783775	ON783790
*Axyris sphaerosperma*_15	Russia, Altai Rep., Kosh-Agach, 1988, Kurbatsky et al. s.n. (LE)	ON775490	ON783761	ON783776	ON783791
*Axyris sphaerosperma*_16	Russia, Tyva Rep., Mongun-Tayginsky distr., 30 July 1981, M. Lomonosova and T. Akimenko 2882 (MW0058743)	ON775491	ON783762	ON783777	ON783792
**Outgroup**					
*Atriplex* *phyllostegia*	U. S. A., Nevada, Churchill Co., Zacharias 992 (UC)	HM005870	HM587590	HM587651	-
*Blitum antarcticum*	Chile, Tierra del Fuego, December 1971, Moore and Goodall s.n. (LE)	MH155315	MH632743	MH152573	MH632745
*Ceratocarpus arenarius*	No voucher	OP550131	OP554752	OP554754	OP554756
*Hablitzia tamnoides*	Bot. Gard. Bonn 3609-90 (BONN), Th. Borschz 3546	AY858590.1	AY270092.1	-	AY858600.1
*Krascheninnikovia lanata*	No voucher	OP550132	OP554753	OP554755	OP554757
*Neomonolepis spathulata*	USA, California, Susanville, August 1983, I.Yu. Koropachinsky et al. 404 (MHA)	MH675518	MH731232	MH152575	MH731230

**Table 2 plants-11-02873-t002:** Primers and cycler programs used for DNA amplification.

Marker	Primer Sequences and Combinations	References	Cycler Program
ITS	ITS5 5′-GGA AGT AAA AGT CGT AAC AAG G-3′	[30]	95 °C for 5 min, 33 cycles of amplification (95 °C for 15 s, 55 °C for 30 s, 72 °C for 40 s), 72 °C for 5 min
	ITS4 5′-TCC TCC GCT TAT TGA TAT GC-3′		
*rbcL* (partial)	rbcLaF 5′- ATG TCA CCA CAA ACA GAG ACT AAA GC-3′	[31]	95 °C for 5 min, 35 cycles of amplification (95 °C for 10 s, 55 °C for 30 s, 72 °C for 40 s), 72 °C for 5 min
	rbcLaR 5′-GTA AAA TCA AGT CCA CCR CG-3′	[32]	
*atpB-rbcL* spacer	atpB-rbcL F 5′-GAA GTA GTA GGA TTG ATT CTC-3′	[33]	95 °C for 5 min, 35 cycles of amplification (95 °C for 20 s, 56 °C for 30 s, 72 °C for 60 s), 95 °C for 20 s, 56 °C for 80 s, 72 °C for 8 min
	atpB-rbcL R 5′-CAA CAC TTG CTT TAG TCT CTG-3′		
*trnL-F* spacer	Tab C 5′-CGA AAT CGG TAG ACG CTA CG-3′	[34]	95 °C for 5 min, 35 cycles of amplification (95 °C for 1 min, 50–65 °C [increasing in 0.3 °C per cycle] for 1 min, 72 °C for 4 min), 72 °C for 5 min
	Tab D 5′-GGG GAT AGA GGG ACT TGA AC-3′		
	Tab E 5′- GGT TCA AGT CCC TCT ATC CCC-3′		
	Tab F 5′ATI′ TGA ACT GGT GAC ACG AG 3′		

**Table 3 plants-11-02873-t003:** The coding of the geographic areas of *Axyris* species and outgroups.

Taxon	Geographical Areas
*Axyris amaranthoides*	A
*Axyris caucasica*	C
*Axyris hybrida*	A, B
*Ax* *y* *ris mira*	D
*Axyris prostrata*	A, B, D
*Axyris sphaerosperma*	A, B, E
**Outgroups**	
*Atriplex phyllostegia*	F
*Blitum antarcticum*	F
*Ceratocarpus arenarius*	A
*Hablitzia tamnoides*	C
*Krascheninnikovia lanata*	F
*Neomonolepis spathulata*	F

**Table 4 plants-11-02873-t004:** Results of the biogeographic analysis using BioGeoBEARS.

	LnL	Numparams	d	e	j	AIC	AIC_wt
DEC	−45.76	2	0.010	0.010	0	95.51	0.055
DEC + J	−41.92	3	0.0038	1.0 × 10^−12^	0.15	89.84	0.94
DIVALIKE	−50.19	2	0.010	0.010	0	104.4	0.0007
DIVALIKE + J	−50.14	3	0.010	0.010	0.0001	106.3	0.0003
BAYAREALIKE	−57.6	2	0.75	0.49	0	119.2	4.0 × 10^−7^
BAYAREALIKE + J	−54.08	3	0.42	0.30	0.55	114.2	4.9 × 10^−6^

## Data Availability

All the newly obtained DNA sequences were deposited in GenBank https://www.ncbi.nlm.nih.gov/genbank/ (accessed on 2 June 2022).

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
