# Peer review of "Biogeography and Systematics of the Genus Axyris (Amaranthaceae s.l.)"

_plants, 2022, doi:10.3390/plants11212873_

Round 1

Reviewer 1 Report (Previous Reviewer 1)

I insist with my previous comment on figure 4. As it is is quite difficult to understand.

Author Response

Figure 4 (presumed dispersal events mapped) should be improved, with the distribution of each species represented, not lumping differet species together, which makes it quite difficult to understand.

- We have decided to delete this figure.

Reviewer 2 Report (Previous Reviewer 2)

I reviewed the revised version of the manuscript “Biogeography and Systematics of the Genus Axyris (Amaranthaceae s.l.)” submitted to Plants and I think that this version presents many improvements and I suggest its publication after some minor modifications.

My main concern is related to the section 2.1 in Results (Species Diversity of Axyris in Asia). Actually, all these results concern the geographic distribution of Axyris, and not really diversity in the genus). So, the authors must modify this section according to the results of geographic distribution.

2.2 Molecular Phylogeny and Dating of Axyris: please change this section name for “Dated molecular phylogeny”

Line 75: change “spacers” by “intergenic spacers”

Figures 2 and 3 present a very poor resolution, it should be improved

Figure 4: in line 95 it is write (Figure 4; marked with orange), however, in figure’s caption it is brown. Please correct this.

Discussion

The beginning is very confusing, 3.1 and 3.2 could be presented together.

I suggest to compile all three figures (5, 6 and 7) in only one using distinct symbols to represent each speciesdistribution. Moreover, I´m not in agreement with this presentation in the discussion section. The figures must appear in the results.

3.3. The Origin of Axyris: it is very interesting and I suggest to initiate the discussion here

3.3. Systematics of Axyris: I strongly suggest a presentation of an identification key

Author Response

I reviewed the revised version of the manuscript “Biogeography and Systematics of the Genus Axyris (Amaranthaceae s.l.)” submitted to Plants and I think that this version presents many improvements and I suggest its publication after some minor modifications.

My main concern is related to the section 2.1 in Results (Species Diversity of Axyris in Asia). Actually, all these results concern the geographic distribution of Axyris, and not really diversity in the genus). So, the authors must modify this section according to the results of geographic distribution.

– We have improved this section also using the previous study of Sukhorukov (2011).

2.2 Molecular Phylogeny and Dating of Axyris: please change this section name for “Dated molecular phylogeny” – Done

Line 75: change “spacers” by “intergenic spacers” – Done

Figures 2 and 3 present a very poor resolution, it should be improved – It is only a word file. All figures will be substituted at the final stage.

Figure 4: in line 95 it is write (Figure 4; marked with orange), however, in figure’s caption it is brown. Please correct this. – Done

Discussion

The beginning is very confusing, 3.1 and 3.2 could be presented together.

- I don’t think so. In the section 3.2., I have discussed the geographical puzzle of A. sphaerosperma as a separate section. The section 3.1. describes indeed the distributional improvements of three Axyris in Siberia.

I suggest to compile all three figures (5, 6 and 7) in only one using distinct symbols to represent each speciesdistribution. Moreover, I´m not in agreement with this presentation in the discussion section. The figures must appear in the results.

- A plate of three figures will be too small to recognize all records. Some fragments of three species are overlapping, and this combined figure will not make sense. The figures now appear in the Results.

3.3. The Origin of Axyris: it is very interesting and I suggest to initiate the discussion here. – It could be rather done in other paper, because the data for some other genera are still lacking or confusing. I don’t think that we should start a further discussion. It will be a speculation.

3.3. Systematics of Axyris: I strongly suggest a presentation of an identification key

- I have added a key, but it seems to be redundant in this paper.

Thank you for some valuable suggestions.

Round 2

Reviewer 1 Report (Previous Reviewer 1)

Instead of redoing the figure 4 the authors have decided to delete it, which is OK to me.

This manuscript is a resubmission of an earlier submission. The following is a list of the peer review reports and author responses from that submission.

Round 1

Reviewer 1 Report

The manuscript represents an interesting, well written contribution. Methods, results and discussion are clear and to the pint. My only concern is that the biogeographic hypothesis derived from the DEC analysis should be explained more clearly, and, additionally, figure 4 (presumed dispersal events mapped) should be improved, with the distribution of each species represented, not lumping differet species together, which makes it quite difficult to understand.

Reviewer 2 Report

I reviewed the manuscript “Biogeography and Systematics of the Genus Axyris (Amaranthaceae s.l.)” submitted to Plants and I think that it still deserves more work before publication.

I observed several points that need to be improved at work. The main points are reported below:

The authors considered that Axyris has six species, however Kew (Plants of the World Online) considers seven species. Why A. koreana was not considered in this work? The authors should review the geographic distribution data of the species. Confirm geographic distribution also: according to Kew (Plants of the World Online), A. amaranthoides presents a native range in E. Europe to N. Korea, a wider distribution range comparing to this work.

In my opinion, the methods used to survey species records for biogeographic analyses are not correct. Why don't authors used databases like Gbif, for example? In a quick search, I found 7,794 results for A. amaranthoides.

With regard to the data of DNA sequences, I point out that Table 1 does not contain the genbank codes for the ITS region. In addition, the sequences are not available in genbank as reported by the authors, the codes cited do not have records.

Please, check support indexes between text and trees. Some values mentioned in the text do not correct the respective nodes in the tree.

In item 2.3 all p values are not significant, the authors should justify and discuss these data.

The discussion is very descriptive, they could relate the distribution patterns with other bioclimatic factors that would help correlate such distribution and also other taxa with similar distribution. And the discussion about phylogenetic relationships and proposed dates for diversification points? These results were not discussed.

it is not necessary to cite the figures in the discussion.

With regards to systematics of Axyris, I suggest presenting a dichotomous key to the three species in the section Axyris sect. Hybridae Sukhor., sect. nov.

Reviewer 3 Report

This is a study that reappraises herbarium specimens of the genus Axyris from Siberia, reconfirms the distribution of each taxon, and performs phylogenetic and biogeographical analyses within the genus. The effort involved in reappraising the specimens is mind-boggling.

However, the main result, phylogenetic analysis, is not reliable. The nucleotide sequence data of individuals of taxa used by the authors as outgroups are actually sequences of loci obtained from multiple individuals that are added together for analysis. In other words, the individuals used as outgroups do not exist as real organisms. They are chimeras. Some of the data used has no voucher specimen. The analysis results obtained by using chimaeras as an outgroup are not reliable. It is recommended that the data of the outgroups be re-compiled with real individuals, that the phylogenetic analysis itself be removed from the results, and that the results of the herbaria survey alone be recompiled. 

It is necessary to discuss more how A. sphaerosperma, which has a particularly disjunct distribution during the repeated glacial and interglacial periods, is thought to have expanded in relation to climatic changes.